# Strategies to reduce stigma and discrimination in sexual and reproductive healthcare settings: A mixed-methods systematic review

Meghan A. Bohren[1]*, Martha Vazquez Corona[1], Osamuedeme J. Odiase[2], Alyce N. Wilson[3,4], May Sudhinaraset[5], Nadia Diamond-Smith[2,6], Jim Berryman[7], Özge Tunçalp[8], Patience A. Afulani[2,6]

1 Gender and Women's Health Unit, Centre for Health Equity, School of Population and Global Health, University of Melbourne, Carlton, Victoria, Australia, 2 Institute for Global Health Sciences, University of California, San Francisco, San Francisco, California, United States of America, 3 Maternal, Child and Adolescent Health Program, Burnet Institute, Melbourne, Victoria, Australia, 4 Nossal Institute, School of Population and Global Health, University of Melbourne, Carlton, Victoria, Australia, 5 Jonathan and Karin Fielding School of Public Health, University of California, Los Angeles, Los Angeles, California, United States of America, 6 Department of Epidemiology and Biostatistics, School of Medicine, University of California, San Francisco, San Francisco, California, United States of America, 7 Brownless Biomedical Library, University of Melbourne, Melbourne, Victoria, Australia, 8 Department of Sexual and Reproductive Health and Research, including UNDP/UNFPA/UNICEF/WHO/World Bank Special Programme of Research, Development and Research Training in Human Reproduction (HRP), World Health Organization, Geneve, Switzerland

* meghan.bohren@unimelb.edu.au

**Data Availability Statement:** Data available within the article or its supplementary materials.

## Abstract

Stigma and discrimination are fundamental causes of health inequities, and reflect privilege, power, and disadvantage within society. Experiences and impacts of stigma and discrimination are well-documented, but a critical gap remains on effective strategies to reduce stigma and discrimination in sexual and reproductive healthcare settings. We aimed to address this gap by conducting a mixed-methods systematic review and narrative synthesis to describe strategy types and characteristics, assess effectiveness, and synthesize key stakeholder experiences. We searched MEDLINE, CINAHL, Global Health, and grey literature. We included quantitative and qualitative studies evaluating strategies to reduce stigma and discrimination in sexual and reproductive healthcare settings. We used an implementation-focused narrative synthesis approach, with four steps: 1) preliminary descriptive synthesis, 2) exploration of relationships between and across studies, 3) thematic analysis of qualitative evidence, and 4) model creation to map strategy aims and outcomes. Of 8,262 articles screened, we included 12 articles from 10 studies. Nine articles contributed quantitative data, and all measured health worker-reported outcomes, typically about awareness of stigma or if they acted in a stigmatizing way. Six articles contributed qualitative data, five were health worker perspectives post-implementation and showed favorable experiences of strategies and beliefs that strategies encouraged introspection and cultural humility. We mapped studies to levels where stigma can exist and be confronted and identified critical differences between levels of stigma strategies aimed to intervene on and evaluation approaches used. Important foundational work has described stigma and discrimination in

**Funding:** This research was made possible by the support of the American People through the United States Agency for International Development (USAID) and the UNDP/UNFPA/UNICEF/WHO/ World Bank Special Programme of Research, Development and Research Training in Human Reproduction (HRP), Department of Sexual and Reproductive Health and Research, WHO. (MAB, PA). MAB's time is supported by an Australian Research Council Discovery Early Career Researcher Award (DE200100264) and a Dame Kate Campbell Fellowship. PA's time is partially supported by a Eunice Kennedy Shriver National Institute of Child Health & Human Development Career development award (R00HD093798). AW's time is supported by a National Health & Medical Research Council Postgraduate Scholarship (APP1151585). The funders had no role in study design, data collection and analysis, decision to publish, or preparation of the manuscript.

**Competing interests:** The authors have declared that no competing interests exist.

sexual and reproductive healthcare settings, but limited interventional work has been conducted. Healthcare and policy interventions aiming to improve equity should consider intervening on and measuring stigma and discrimination-related outcomes. Efforts to address mistreatment will not be effective when stigma and discrimination persist. Our analysis and recommendations can inform future intervention design and implementation research to promote respectful, person-centered care for all.

## Introduction

Sexual and reproductive health and rights (SRHR) are essential to achieve equitable and sustainable development [1]; however, persistent inequities remain related to unintended pregnancy, pregnancy and childbirth complications, unsafe abortion, infertility, sexually transmitted infections (STIs), reproductive cancers, and gender-based violence. SRHR are integral components of health, social, and economic development [1], but remain out of reach for many people globally. Political ideologies, social and cultural expectations around gender equality, reproductive choices, and sexuality continue to threaten SRHR [1]. The 2030 Sustainable Development Agenda highlights two targets specific to achieving universal access to sexual and reproductive health, health-care services and reproductive rights (targets 3.7 and 5.6) [2]. While this ambitious agenda acknowledges universal access, more work is needed to ensure an equitable approach to SRHR–an approach that ensures that the needs and priorities of specific groups of people, who are persistently disadvantaged by existing systems of power, are not left behind.

### Stigma and discrimination in healthcare settings

Stigma and discrimination manifest in broader society and in healthcare settings, and reflect levels and types of privilege, power, and disadvantage within society. Stigma and discrimination are related concepts with distinct differences [3]. Link and Phelan (2001) define stigma as the co-occurrence of "labeling, stereotyping, separation, status loss and discrimination" in contexts where power is exercised [4]. Discrimination, on the other hand, is defined as unfair and unjust actions towards an individual or group based on real or perceived status or attributes, medical conditions, socioeconomic status, gender, race, sexual identity, or age. Discrimination is a fundamental feature and expression of stigma, occurring both at the structural-level (societal conditions constraining opportunity or well-being) and individual-level (unequal or unfair treatment based on membership of a social group) [3]. Link and Phelan describe discrimination as the endpoint of the stigmatization process [4], while others view discrimination as a manifestation of the stigmatization process [5]. Both perspectives likewise place stigma and discrimination as fundamental causes of health inequities for three key reasons [3, 6, 7]: 1) stigma and discrimination influence several health outcomes through multiple pathways; 2) stigma and discrimination limit access to resources that can be used to avoid or minimize health risks or consequences; and 3) stigma and discrimination are related to health inequalities irrespective of time or place. Central to the conceptualization of stigma is the context and experience of power, privilege, and dominance that fosters environments of oppression and 'othering' of those who are stigmatized or discriminated against [8, 9]. When stigma or discrimination is experienced in healthcare settings, it is a violation of human rights [10, 11].

   Stigma and discrimination are multi-level conditions that can differ across contexts, but there are common drivers, manifestations, and consequences present across settings and

populations [12], including in healthcare settings. In Fig 1, we depict a multi-level stigma model that reflects these complexities within SRHR settings, mapped across the levels of internalized stigma, perceived stigma, enacted stigma, structural stigma, and layered stigma [13]. Internalized stigma refers to when a person with certain attributes is aware of public stigma about their attributes, agrees with these stereotypes, and applies the stigma to themselves [13]. Perceived stigma reflects an individual's awareness of public stigma or beliefs that others hold stigmatizing thoughts about their condition or group [14, 15]. Enacted stigma refers to the manifestations of unfair treatment arising from adverse social judgment [16]. Structural stigma encompasses the societal conditions, sociocultural norms, and institutional policies that influence the opportunities and well-being of stigmatized groups [17].

Reproductive justice, a movement led by women of color in the United States, strives to achieve health equity, end discrimination and oppression, and challenge medical hierarchies through the recognition that the intersectionality of social, political and economic identities shape peoples' abilities to access safe, appropriate, and respectful SRHR services [19, 20]. Through the lens of reproductive justice, we can better understand the intersecting influences of laws and social policies on people's and communities' health and well-being, and how these laws and policies contribute to longstanding injustices and structural oppression. Tantamount to achieving reproductive justice is acknowledging, then dismantling, oppressive systems that prohibit all people from achieving their fullest potential. Reproductive justice, therefore, can extend conceptualizations of stigma and discrimination from the interpersonal level, to better implicate the institutions and systems that perpetuate stigma and discrimination at the structural level. Similarly, global evidence has shown that women are mistreated during childbirth, which includes both structural and interpersonal discrimination while seeking care [21]. Eliminating stigma and discrimination within healthcare settings is critical to the attainment of reproductive justice and respectful care for all.

Structural and individual experiences or consequences of stigma and discrimination within sexual and reproductive healthcare have been well documented. Institutional and broader health system and social policies may drive stigma and discrimination in healthcare settings [12], where manifestations of stigma are both overt and covert [12, 21, 22]. Likewise, health workers' perceptions, fears, belief systems, negative attitudes, moral distress, and lack of awareness about stigma, of a health condition, or the population may contribute to stigma in SRHR settings [12]. Individuals who are disadvantaged by systems of power and seeking sexual and reproductive health services may receive unfair or unequal treatment or have worse health outcomes compared to those who are in more privileged positions. Research has consistently shown that individuals who experience stigma and discrimination in healthcare settings may delay or forego seeking healthcare in the future [23]. Once individuals from stigmatized groups access the health system, they may be denied care or experience delays in receiving care. They may also be provided lower-quality care, receive care that is not culturally appropriate, or experience mistreatment such as discrimination, verbal and physical abuse, and denial of care [12, 24, 25]. Collectively, this can result in dissatisfaction and loss of trust in the health system, resulting in a vicious cycle of delaying care or not seeking care at all [26]. Additionally, it can lead to delayed diagnoses and initiation of care, as well as lower adherence and engagement in care, leading to poor health outcomes [27, 28]. The experience of discrimination itself has profound impacts on people's physical and mental health as well as their general wellbeing [29–31].

Layered stigma refers to the interaction between multiple stigmatized identities within an individual or group [18]. For example, in the United States, persistent disparities exist in maternal mortality where Black women are three times more likely to die than non-Hispanic white women, and these experiences occur within a history of racist reproductive policies [32].

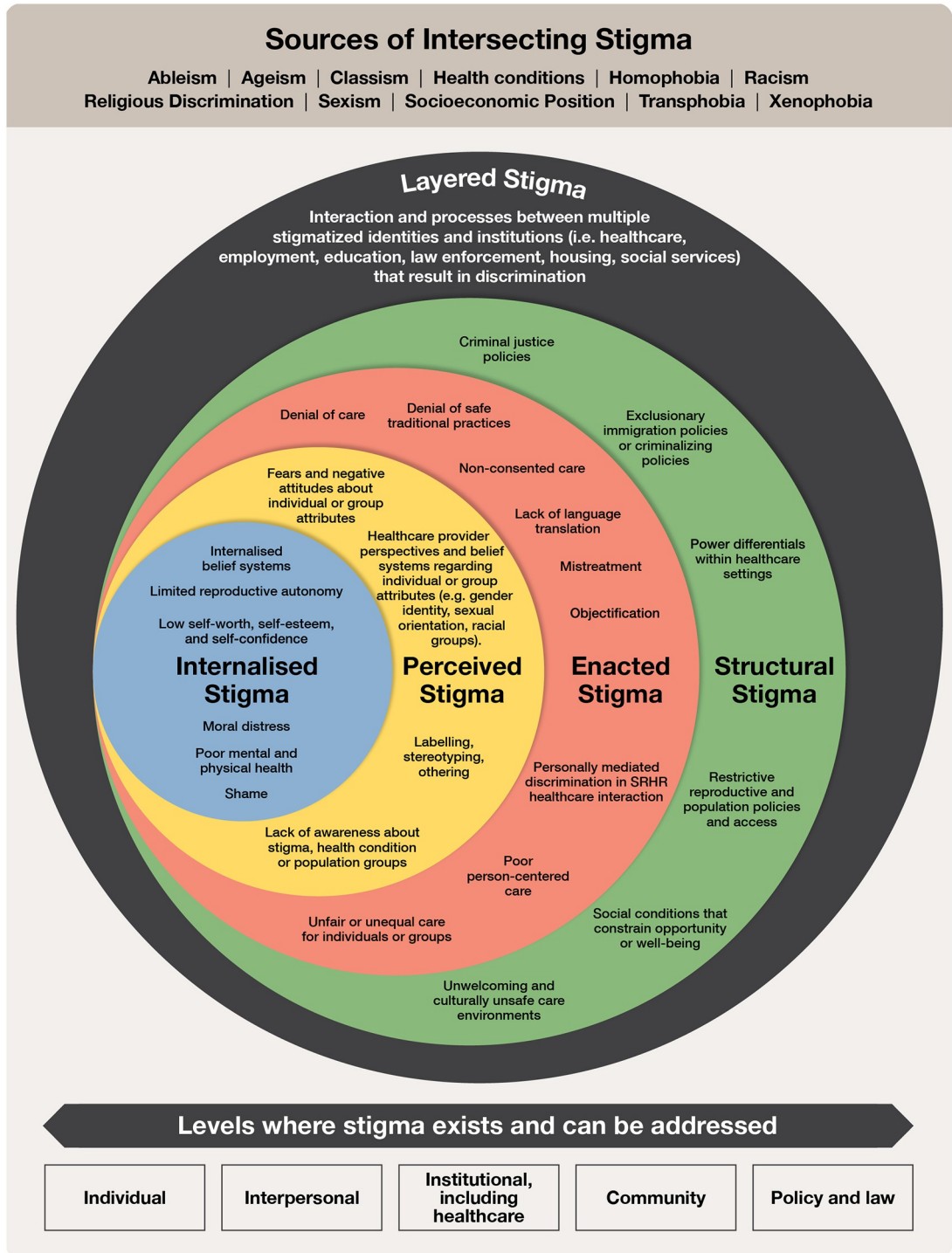

**Fig 1. Multi-level stigma model for sexual and reproductive health and rights.** This figure depicts the complexities of where stigma and discrimination exist within sexual and reproductive health and rights, and the levels where it can be confronted. Footnote: [i] Internalized stigma: person with certain attributes is aware of public stigma about their attributes, agrees with these stereotypes, and applies the stigma to themselves [13] [ii] Perceived stigma: individual's awareness of public stigma or beliefs that others hold stigmatizing thoughts about a condition [14, 15] [iii] Enacted stigma: manifestations of unfair treatment arising from adverse social judgment [16] [iv] Structural stigma: societal conditions, sociocultural norms, and institutional policies that influence the opportunities and well-being of stigmatized groups [17] [v] Layered stigma: interaction between multiple stigmatized identities within an individual or group [18].

These reproductive policies intentionally practiced non-consensual or involuntary sterilization on Black women as a way of limiting the population of the perceived inferior group [33], and are a clear representation of structural racism. In Australia, racist government policies forcibly removed babies and children from Aboriginal families as part of the white Australia 'assimilation policy', leading to profound intergenerational trauma, loss of identity, and grief [34]. During apartheid South Africa, the Afrikaner government racialized and weaponized family planning by providing tax incentives for white women to procreate, while simultaneously promoting contraception for Black and Coloured women to limit fertility [35]. Present-day disparities are likewise driven by racism, which manifests in various ways including Black and Indigenous women lacking access to sexual and reproductive services, abuse in health care settings, assaults on migrant women's reproductive autonomy, including forced sterilizations in detention centers, and the ongoing stress of detrimental colonial processes and living in race-conscious societies [36–40].

Stigma and discrimination also drive inequities in sexual and reproductive outcomes based on people's social status, including socioeconomic status, marital status, and age. Prior studies have shown women of low socioeconomic status (measured variably by household wealth, caste, education, literacy, and employment status) are more likely to have poorer experiences during childbirth than those of higher socioeconomic status [41–43]. A World Health Organization (WHO) study on the mistreatment of women during childbirth in Ghana, Guinea, Myanmar and, Nigeria showed that adolescents and younger women were twice as likely to be physically abused and four times as likely to be verbally abused during childbirth, compared to older women, and 3% of women reported experiencing stigma or discrimination [22, 44]. Other forms of stigmatization based on gender and sexual orientation, disability, disease conditions such as HIV, STIs, tuberculosis, leprosy, substance use, and mental illness, as well as the type of care such as infertility and abortion services have been extensively documented as a barrier to care [24, 26, 41, 45–47]. Furthermore, the use of certain health services can be stigmatizing in and of themselves (such as abortion, STI, and HIV services), and thus can result in stigma and discrimination of healthcare users, providers, and communities.

## Strategies to reduce stigma and discrimination in healthcare settings

Given the multi-level drivers and manifestations of stigma and discrimination, sustainable and scalable strategies to reduce stigma and discrimination in sexual and reproductive healthcare settings likely need to reflect this complexity. To date, much of the literature on strategies to reduce stigma and discrimination have been in the HIV field, with comparably less research on sexual and reproductive health. Nyblade and colleagues (2019) synthesized evidence on facility-based interventions to reduce stigma in HIV, mental illness, and substance abuse and identified six main approaches to reduce stigma broadly classified in Box 1 [12]. Many interventions identified had multiple components to reduce stigma [12], and the intervention mechanisms of action can be summarized as follows: improving healthcare provider awareness of stigma and contact with the stigmatized group may translate into improved practices of empathy in clinical encounters. Training healthcare providers on concrete tools and approaches to address stigma and work with stigmatized groups transforms abstract theories of stigma into concrete action in how they can provide better care. Policy reform may create more enabling environments for safe, respectful, inclusive, and anti-racist care without creating further barriers to certain groups seeking care. Lastly, strengths-based and community-driven approaches can empower individuals and groups who experience social, cultural, and economic oppression to achieve their SRHR by drawing on their own strengths and assets to lead initiatives or programs to tackle stigma and discrimination.

> ### Box 1. Six approaches to reduce stigma and discrimination in healthcare settings.
>
> This figure six approaches to reduce stigma and discrimination in healthcare settings (adapted from Nyblade and colleagues [12]).
>
> 1. **Provision of information** to teach healthcare providers about health conditions, or stigma manifestations and consequences
>
> 2. **Skills-building** for healthcare providers to improve competence in working with a stigmatized group
>
> 3. **Participatory learning** for either/both healthcare providers and healthcare users to engage in the intervention
>
> 4. **Contact with stigmatized groups** to humanize the stigmatized group and encourage healthcare providers to develop empathy
>
> 5. **Empowerment approach** to improve healthcare user coping mechanisms to overcome stigma
>
> 6. **Structural or policy reform** to improve or create redress mechanisms, or facility restructuring

While perceptions, experiences and impacts of stigma and discrimination in health have been increasingly well-documented, a critical gap remains in understanding the evidence on strategies to reduce stigma and discrimination in sexual and reproductive healthcare. We aim to address this gap by conducting a systematic review of strategies (inclusive of interventions, programs, and policies) to reduce stigma and discrimination in sexual and reproductive healthcare settings, in order to describe types and characteristics of strategies, assess effectiveness, and synthesize key stakeholder experience and perceptions of the strategies.

## Methods

This is a mixed-methods systematic review and narrative synthesis of strategies to reduce stigma and discrimination in sexual and reproductive healthcare services globally. The review is reported according to the Preferred Reporting Items for Systematic Reviews and Meta-Analyses (PRISMA) statement [48] (S1 Table), Cochrane Effective Practice and Organisation of Care guidance [49], and the protocol is available (S1 Text) and registered on the International prospective register of systematic reviews ((PROSPERO): CRD42020221054).

### Types of studies

For the quantitative component, randomized and non-randomized trials, pre-post studies (with or without a control group), interrupted time series, and other designs that compare the strategy to reduce discrimination with usual care were eligible for inclusion. Our operational definition of what constitutes a 'strategy' was inclusive of approaches such as policy reform, introducing new models of care, healthcare provider training (workshops, sensitization training, simulation), mystery clients, and community-based approaches targeting healthcare users.

Studies published in abstract form were not eligible for inclusion, unless additional information could be obtained from study authors. Where we identified study protocols, we forward-reference searched to identify any results publications for relevance. We included studies that focused on strategies with the following characteristics: 1) studies that directly aimed to reduce stigma or discrimination in SRHR, or reducing inequity or promoting equity by reducing stigma and discrimination in SRHR; 2) studies that included a quantitative outcome related to SRHR and stigma or discrimination (e.g. experiences of stigma or discrimination, provider perceptions of stigma and discrimination); and 3) strategies that targeted healthcare users or communities, providers, health facilities or systems, health laws, or policies.

For the qualitative component, primary studies that used qualitative or mixed-methods designs to evaluate user or provider experiences of strategies to reduce stigma or discrimination were eligible for inclusion (e.g. process evaluations, ethnographies, case studies, phenomenological studies). Studies were eligible for inclusion regardless of whether they were conducted alongside studies of effectiveness (sibling studies) and did not need to have a comparison group to be eligible.

## Topics of interest, types of participants, and settings

We included quantitative, qualitative, and mixed-methods studies that focused on strategies to reduce stigma and discrimination in sexual and reproductive healthcare settings. We have defined sexual and reproductive healthcare settings for this review as the following services, based on the Guttmacher–Lancet Commission on Sexual and Reproductive Health and Rights [1]:

- **Maternal health services**, including preconception care (pregnancy and infertility testing, counselling, and services), antenatal care (including STI screening and treatment, and preventing mother-to-child transmission of HIV (PMTCT)), childbirth care, and postpartum mother and baby care up to six weeks after birth (including immunization, breastfeeding)

- **Contraceptive counselling and services**, including STI screening

- **Safe abortion and post-abortion services**

- **Reproductive tract cancers including cervical cancer counselling and services**

- **Gender-based violence screening**

- **Infertility testing and treatment**

We excluded STI and HIV counseling, testing, and treatment services if conducted outside the context of contraceptive or maternal health services, as these topics have been well documented in the literature [12, 27, 28]. We excluded 'comprehensive sexuality education' as this is typically delivered in school-based settings, and the setting of interest in this review is healthcare settings. We adopted an intersectional approach to understanding discrimination by including studies that explore discrimination based on (but not limited to) race, ethnicity, Indigenous identity, social status, gender, sexuality, dis(ability), age, religion, migration or visa status, and the intersections between these identities.

Studies that included perspectives of healthcare users, family members of healthcare users, community members, healthcare providers, policy-makers, or other key stakeholders were eligible for inclusion, with no restrictions to sociodemographic characteristics or identity. Studies conducted in any country globally, and in any type of setting where sexual and reproductive healthcare is received (e.g. health facilities, community-based care, home-based care, or other types of institutional-based care) were eligible.

## Search methods and study selection

We searched the following databases from inception to date of search, without any limits on language or publication date (S2 Text): MEDLINE (April 6, 2022), CINAHL (March 29, 2022), and Global Health Ovid (April 13, 2022). In addition to database searching, we reviewed the reference lists of included studies, conducted a forward citation search of all included studies using Web of Science, and conducted a grey literature search using OpenGrey (www. opengrey.eu). We collated all citations identified from different searches into Covidence (Covidence systematic review software, Veritas Health Innovation, Melbourne, Australia, www.covidence.org) and removed duplicates. Two review authors independently assessed each record for potential eligibility, and excluded references that did not meet the eligibility criteria (MAB, PAA, MVC, OJO, ANW). Two independent reviewers assessed full texts of potentially eligible studies, and disagreements were resolved through discussion and consensus with a third reviewer (MAB, PAA, MVC, OJO, ANW). For title and abstract assessment of studies published in languages that none of the review team are fluent in (languages other than English, Spanish, or French), we carried out initial translation through Google Translate. If the translation indicated agreement with the inclusion criteria, or if the translation is insufficient to decide, we consulted other colleagues in our networks to assist in assessing full text for inclusion.

## Data extraction and critical appraisal

We designed a data extraction form for this review to extract data on study setting, sample characteristics, objectives, guiding frameworks, study design, strategy design and components, data collection tools and analysis methods, and author conclusions. For quantitative studies, we extracted data on the outcomes of interest related to discrimination (e.g. percentages, odds ratios, relative risks, prevalence estimates). For qualitative studies, we extracted the qualitative findings including author themes and participant quotations. Two independent reviewers critically appraised the methodological limitations of included studies using the Cochrane ROBINS-I tool [50] for quantitative studies and an adaptation of the Critical Appraisal Skills Programme (CASP) tool (www.casp-ul.net) for qualitative studies (MAB, PAA, MVC). Disagreements were resolved through discussion or involving a third review author where necessary.

## Data management, analysis and synthesis

We used a narrative synthesis approach, which is a particularly useful approach when analyzing data from different types of studies [51]. A narrative synthesis approach focuses on the interpretive synthesis of the narrative findings of the research to sort and analyze studies into more homogenous groups based on critical components such as study design, study setting, program components, types of participants, type of health service and health topic, types of outcome, and direction or magnitude of effect [51]. There are four key elements of narrative synthesis recommended for implementation-focused narrative reviews, and these elements are analyzed iteratively throughout the review and synthesis process [51]. First, we developed a preliminary synthesis, which consisted of an initial textual descriptive analysis of included intervention studies and their findings. This preliminary synthesis allowed us to identify and evaluate initial factors, components, and processes that affected the construction of analytic outputs from the subsequent analysis steps. We used this textual descriptive analysis to group and cluster extracted data based on similar features, such as type of strategy, study location, and context, whether the strategy was designed with or by the stigmatized group, and type of participants (Table 1). Second, we explored the relationships between and across studies to

**Table 1. Characteristics of included studies.**

| Author (year) | Aim of study | Health area | Setting | Study design | Description of participants | Description of intervention | Topics covered in intervention | Type of stigma aiming to address |
|---|---|---|---|---|---|---|---|---|
| **Duby 2019** [62] | To evaluate the effects of the integrated key population sensitization training intervention for healthcare workers. | Sexual health and HIV | South Africa: 5 provinces received training, 2 provincial capitals evaluated (Bloemfontein, Free State and Mafikeng, Northwest Province) | Pretest-posttest single group design in a mixed-methods evaluation (survey and IDIs before and after the intervention) | Intervention participants: unspecified number of health workers who participated in the training Evaluation participants Quantitative: 401 health workers completed the survey at baseline & 405 at endline) Qualitative: 8 health workers participated in IDIs at baseline and 3 endline | 2-day sensitization training for health workers using training manual and facilitation guide for the 'Integrated Key Populations Sensitivity Training Programme for Healthcare Workers in South Africa'. Followed up with 1-day training on stigma 5–6 months following. | Social norms and values, human sexuality, sexual behavior, legal and rights context, socio-structural marginalization and prejudice, interventions to foster enabling healthcare environment | Perceived stigma |
| **Geibel 2017** [60] | To assess the effects of the stigma reduction trainings on service provider attitudes, and young client satisfaction with services | Sexual health and HIV | Bangladesh: 270 Marie Stopes Bangladesh health service facilities in 38 target districts | Pretest-posttest single group design (surveys at 3 time points: before intervention, and 6- and 12-months post-intervention | "Link up" service providers (300), and service users aged 15–24 from "at risk" populations (HIV+, sexually active and unmarried youth, sex workers, men who have sex with men, transgender people) (survey 1: 264, survey 2: 371) | 2-day HIV and SRHR training with a 90 min session on stigma and gender issues + 1 day supplementary training on stigma after 6 months of initial training | HIV, SRHR, gender, naming stigma, experiences of stigma, key populations stigma, values clarification, talking about sex, sexuality, men who have sex with men, transgender, challenging stigma, stigma-free services | Perceived & enacted stigma |
| **Harris 2011, Martin 2014, Debbink 2016** [54–56] | Harris 2011: To provide a safe space for abortion providers to discuss their experiences and evaluate an intervention designed to ameliorate stigma's burdens. Martin 2014: To explore how abortion providers compare to others in helping professions with regards to their professional quality of life, and if scores on the Abortion Provider Stigma Survey (APSS) and the Professional Quality of Life (ProQOL) scale change over time? Debbink 2016: To describe the theory, development and implementation of the Providers Share Workshops. | Abortion | USA: Workshops facilitated at 7 abortion clinics (Martin 2014, Debbink 2016) Workshops facilitated at 1 abortion clinic (Harris 2011) | Pretest-posttest single group design in a mixed-methods evaluation (surveys at 3 time points: before intervention, 3 weeks post-intervention and 12 months post-intervention; IDIs post-intervention) (Martin 2014: surveys at 3 time points; Harris 2011: qualitative; Debbink 2016: qualitative and surveys) | 17 female health workers providing abortion services (doctors, managers, nurses, counselors, surgical assistants) participated in the workshops (Harris 2011) 69 health workers participated in the survey (Martin 2014) 79 health workers participated in the workshops (Debbink 2016) | 6 sessions in which abortion providers meet to explore their experiences providing abortion care in a "Provider Share" workshop. Topics included: 1) What abortion work means to me; 2) Memorable stories; 3) Abortion and identity; 4) Abortion politics; 5) Future directions for self-care; and 6) Reflections on the Workshop. | Meaning of abortion work, memorable stories, identity, politics, self-care, reflections | Internalized stigma |

*(Continued)*

**Table 1.** (Continued)

| Author (year) | Aim of study | Health area | Setting | Study design | Description of participants | Description of intervention | Topics covered in intervention | Type of stigma aiming to address |
|---|---|---|---|---|---|---|---|---|
| **Jadwin-Cakmak 2020** [59] | To describe and evaluate the Health Access Initiative, an intervention to improve the general and sexual health care experiences of sexual and gender minority youth. | Sexual health | USA: 10 sites in Southeast Michigan, including 3 health departments, a school-based health clinic, 2 community health centers, 2 youth-specific health centers, 1 pediatric clinic, and 1 HIV clinic | Pretest-posttest single group design in a mixed-methods evaluation(surveys and IDIs at before and after the intervention) | 101 participants completed online training; 153 participants completed in-person training (doctors, physician's assistants, nurses, medical assistants, social workers, psychologists, administrators, health educator, community health workers | 1-hour online training and 2-hour in person training, followed by 3 months of site-specific technical assistance to improve clinic- and structural-level issues related to LGBTQ + care. The online training | LGBTQ+ sexuality, gender, identity, cultural humility, stereotypes, patient-centered care, technical assistance to develop resources, policies and procedures | Structural, enacted & perceived stigma |
| **Kinn 2003** [63] | To evaluate client and staff views on existing facilities and services, before and after the convergence of genitourinary medicine, family planning and women's health. | SRH | UK: 3 health service settings (family planning, genitourinary medicine and women's health) in Glasgow | Pretest-posttest single group design in a mixed-methods evaluation(surveys and IDIs before and after the intervention) | Client surveys: pre (1335) and post (644) Staff surveys: pre (88) and post (77) Qualitative interviews with staff interviews: pre (83) and post (89) | Service reconfiguration to merge family planning, genitourinary medicine, and women's health services in order to provide integrated services using a social model of health. | Service reconfiguration and impacts on joining services, confidentiality, training, stigma. | Structural stigma |
| **Littman 2009** [57] | The purpose of this study is to get feedback on the proposed intervention of introducing abortion patients to a "culture of support". | Abortion | USA: Mount Sinai Family Planning Panel, New York City | Post-only qualitative design (IDIs after intervention) | 22 women who had an abortion at the study clinic and experienced the intervention. | The goal of the intervention was to introduce abortion patients to a "culture of support" by providing validating messages and information about groups and services that support women in their reproductive decisions, addressing stigma, and providing information to help women identify and avoid sources of abortion misinformation. The intervention consisted of a DVD (documentary of women's abortion experiences including stigma), a brochure (validating messages about reproductive decisions and support services) and a discussion (about stigma and negative messaging). | Abortion stigma, secrecy, reproductive choices, support services | Internalized & perceived stigma |

(*Continued*)

**Table 1.** (Continued)

| Author (year) | Aim of study | Health area | Setting | Study design | Description of participants | Description of intervention | Topics covered in intervention | Type of stigma aiming to address |
|---|---|---|---|---|---|---|---|---|
| **Maclean 2018** [61] | To describe the development, content and evaluation of knowledge translation resources and training workshops designed to equip health and social service professionals with the knowledge and skills needed to provide more respectful and inclusive sexual health, harm reduction and sexually-transmitted and bloodborne infection services. | Sexuality, substance use, STBII | Canada: Canadian Public Health Association | Post-test only design (survey after intervention) | 483 health and social services providers | 4 knowledge translation resources were developed: 1) self-assessment tool for providers to reflect on personal attitudes and beliefs, 2) service provider discussion guide to describe communication strategies service providers can use to foster inclusivity and respect, 3) organizational assessment tool with questions to help assess strengths and challenges relevant to providing inclusive health, 4) training workshops to increase awareness and adoption of stigma reduction strategies. | Attitudes and beliefs about STBBIs and stigma, respectful and inclusive care, stigma reduction, privacy, confidentiality | Perceived & enacted stigma |
| **Mosley 2020** [65] | To adapt the Providers Share Workshop content, structure and evaluation tools for a pilot study in 5 sub-Saharan African and Latin American countries. | Abortion | 3 African and 3 Latin American countries [specific countries not reported] | Pretest-posttest single group design (surveys at 3 time points: before the intervention and 2 time points after the intervention) | 152 health workers providing abortion services | Adaptation of the Providers Share Workshop (Harris 2011, & Martin 2014, Debbink 2016) with partner organizations. Two-day retreat with sessions on "What abortion work means to me," "Managing stigma: the decision to disclose," "What abortion work means to my community," "Memorable cases and difficult complications" and "Looking toward the future." | Meaning of abortion work (self and community), managing stigma, memorable stories, future reflections | Internalized stigma |
| **Phiri 2019** [64] | To explore the impact of Umoyo mother-infant pair clinics on retention of HIV-exposed infants in PMTCT care at 12 months after birth. | PMTCT | Zambia: Lusaka and Eastern provinces, at the 28 Umoyo Mother-Infant Pair clinics | Cluster randomized controlled trial (surveys at 3 time points: before the intervention and 2 time points after the intervention) | HIV-exposed infants and HIV+ mothers | The Umoyo clinic is a designated clinic day for HIV+ mothers and their babies to receive routine child health care and ART services including: 1) enhanced, group-based sensitization and intensified Information Education and Counseling, 2) integrated services including HIV and tuberculosis screening, provision of isoniazid TB prophylaxis; family planning, immunizations and ART services, and 3) active defaulter tracing by lay counselors. | Mentorship, integrated HIV services, early infant diagnosis, family planning, improved documentation | Structural & internalized |

(*Continued*)

**Table 1.** (Continued)

| Author (year) | Aim of study | Health area | Setting | Study design | Description of participants | Description of intervention | Topics covered in intervention | Type of stigma aiming to address |
|---|---|---|---|---|---|---|---|---|
| **Seybold 2014** [58] | To summarize a conference intervention that addresses provider bias, and to evaluate impact on attitudes and knowledge of health care providers about substance abuse working with substance-using pregnant patients. | Maternal health & substance use | USA: West Virginia– Charleston Area Medical Center | Post-test only design (survey after intervention) | 70 participants (health workers) at a conference on substance use and health | Workshop facilitated at a conference for health workers on the Transdisciplinary Foundations from the Substance Abuse and Mental Health Services Administration Competencies Model. The workshop was a mix of education on disease concept of addiction, sharing personal stories of addiction, clinical strategies to care for pregnant people with addiction, developing rapport with the patient through role play, and Q&A session on 'finding compassion in your frustration' including how gender bias impacts care. | Understanding addiction, treatment knowledge, application to practice, professional readiness and provider bias | Perceived & enacted |

understand how critical study design and intervention design factors may have influenced the likelihood of implementation success. Key relationships of interest included the relationships between study designs, levels of engagement, and magnitude and directionality of key findings (Table 2). Third, we used a thematic analysis approach [52] to synthesize the qualitative evidence of key stakeholder perspectives and experiences of the strategies. This included line-by-line coding of findings from primary studies and organization into descriptive and analytic themes and interpretations [53]. Fourth, we developed a model to map the strategy aims and outcome measurement across the different levels where stigma can be confronted, reflected in Fig 3. We assessed the robustness of the synthesis using multiple iterative methods to reflect on the methodological quality of the primary studies included in the synthesis and the trustworthiness of our analysis [51]. As described above, we assessed the methodological limitations of included studies using different tools appropriate for different study designs, and considered throughout the analysis process how to minimize bias, for example by ensuring that studies of equal technical quality are given equal weighting, and by clearly stating eligibility criteria across each step of the review (S2 and S3 Tables).

Throughout all stages of this review, we practiced critical reflexivity both as individuals and as a review team, which contributed to improving the robustness of the synthesis. This enabled us to consider, acknowledge and reflect on how our own lived experiences, employment, training, perspectives on discrimination and sexual and reproductive health services, and other factors shaped and influenced how we designed and conducted the review, synthesized, and interpreted the findings. We noted at the start of this review that the strength of our team comes from our own diversity: we have professional expertise and experience in sexual and reproductive health, public health, social sciences, medicine, obstetrics, epidemiology, social and reproductive justice, global health, and First Nations health. We are from different racial, ethnic, and religious backgrounds, and some of us are first- or second-generation migrants.

**Table 2. Quantitative measurement approach outcome evaluation (among n = 9 studies with quantitative outcomes).**

| Author (year) | Measurement approach & tool | Validated tool? | Type of stigma outcome measures | Effect measures used | Impact |
|---|---|---|---|---|---|
| **Duby 2019** [62] | Baseline and endline survey assessing previous training and experience working with key populations, knowledge, attitudes & beliefs about key populations health and behavior | No | Perceived stigma | Percentages and p-values | • Increased provider awareness of the psychosocial vulnerabilities of key populations (violence, stigma, and lack of healthcare access)<br>• Increased awareness that key populations do not access healthcare services due to fear of abuse by health workers<br>• Participants more comfortable providing health services to key populations |
| **Geibel 2017** [60] | Baseline, midterm and endline provider survey (experience with people living with HIV, workplace & personal drivers of stigma and discrimination)<br>Baseline and endline client exit survey (satisfaction with overall services, quality and professionalism) | Provider survey: yes (HIV stigma measurement tool for health facility staff) Client exit survey: no | Perceived & enacted stigma | Percentages and p-values | • Provider questionnaire: increase in reporting that facility has policies protecting HIV+ people from discrimination, belief that they would be disciplined for violating these policies, and provider willingness to provide services to HIV+ people. Decrease in negative attitudes towards HIV+ people.<br>• Client exit interview: Clients more likely to discuss being a member of a key population group and disclosing sexual activity to providers; enacted stigma (feeling provider acted discriminatorily) decreased. |
| **Jadwin-Cakmak 2020** [59] | Baseline and endline provider survey: assessing training's relevance, applicability, intention to apply learnings, and knowledge, attitudes and practices in ability to interact with key populations. | No | Perceived & enacted stigma | Percentages, p-values, and effect size Cohen's d | • Satisfaction with training usefulness, relevance and impact.<br>• Increases in overall knowledge, attitudes and practices scores from baseline to follow up<br>• Increased knowledge from baseline to follow up, decrease in assumptions about patient sexuality, and gender identity.No change in assumptions about use of patients' preferred names or pronouns<br>• High scores on overall perception of a positive environment for key populations |
| **Kinn 2003** [63] | Baseline and endline client questionnaire: views on accessibility, facilities, attitudes of staff, quality of services, and access<br>Baseline and endline provider survey: standard of current services, how integrated services may affect client support. | No | Perceived & enacted stigma | Percentages | • Client survey: Increase in sufficient confidentiality and happiness with overall care.<br>• No difference in quality of service, availability or wait times after integration. Provider survey: Increased quality of standard of services. Mixed opinions on desired level of service integration. |
| **Maclean 2018** [61] | Endline provider surveys: evaluate what they learned, applicability of content, and knowledge. | No | Perceived stigma | Percentages | High agreement (>87%) for increased awareness of forms of stigma, comfort in discussing issues (sexuality, substance use, harm reduction), organizational strategies to reduce stigma, and ability to integrate workshop learnings to practice |

(*Continued*)

**Table 2.** (Continued)

| Author (year) | Measurement approach & tool | Validated tool? | Type of stigma outcome measures | Effect measures used | Impact |
|---|---|---|---|---|---|
| **Martin 2014** [55] | Baseline and endline provider survey: demographics, Professional Quality of Life (ProQOL) scale, Ways of Coping survey, Process subset of Workgroup Characteristics Measure, People and Organizational Culture Profile, and Abortion Providers Stigma Scale | Yes | Internalized stigma | Beta coefficients from multilevel linear regression models | • Abortion providers report higher compassion satisfaction and lower burnout than other healthcare workers<br>• Decrease in experiences of stigma over time<br>• Experiencing abortion stigma is a predictor of lower compassion satisfaction, higher burnout and higher compassion fatigue. |
| **Mosley 2020** [65] | Baseline and endline provider survey: demographics, level of experience in abortion care, level of abortion provider stigma, abortion attitudes, legal safety and advocacy, and provider burnout | Yes | Internalized stigma | Mean scale scores, beta coefficients from linear regression | • Decrease in total abortion stigma<br>• Decreases in abortion caregivers' sub-score for disclosure management and internalized states<br>• Increased perception of legal safety and engaging in advocacy<br>• Decreases in provider burnout |
| **Phiri 2019** [64] | Baseline and endline patient & facility register: HIV-exposed baby retention in care at 12 months (structural) Baseline and endline client survey: perceived social support and stigma of HIV+ mothers (perceived, internalized) | Patient & facility register: no Client exit interview: yes (Social Provisions Scale, HIV/AIDS Stigma Instrument) | Structural, perceived, internalized | Percentages (confidence interval and p-values), difference in difference over time, adjusted odds ratios | • No effect on baby retention in care at 12 months or from 6 to 12 months<br>• Unweighted analysis: no effect on mother Social Support Score, enacted stigma, perceived stigma from healthcare workers or internalized stigma scores.<br>• Sensitivity tests: increase in social support and reducing perceived stigma from healthcare workers |
| **Seybold 2014** [58] | Endline quantitative evaluation (survey) with retrospective pretest surveys: knowledge and confidence in substance abuse management. | No | Perceived | Mean scale values and mean difference | • Increase in knowledge about gender differences in substance abuse.<br>• No effect on other knowledge items or in confidence in skills. |

We currently work at academic institutions, and at organizations that provide patient and community care in high-resource settings, and our projects regularly engage with people who are disadvantaged by existing systems of power at both global and local levels.

## Results

Out of 8,262 articles screened, 12 articles from 10 studies met inclusion criteria (three articles were from one study [54–56]). Fig 2 presents the PRISMA diagram of inclusions.

### Description of included studies

Table 1 describes the characteristics of included studies. Half of the included articles (6/12) were conducted in the United States [54–59], one study conducted in each of Bangladesh [60], Canada [61], South Africa [62], United Kingdom [63], and Zambia [64], and one multi-country study conducted in three sub-Saharan African and three Latin American countries (countries not specified) [65]. Despite no date limits on the search, all included articles were published between 2003 and 2020. The topics within sexual and reproductive health varied substantially in the included articles: five were on abortion [54–57, 65], two on sexual health

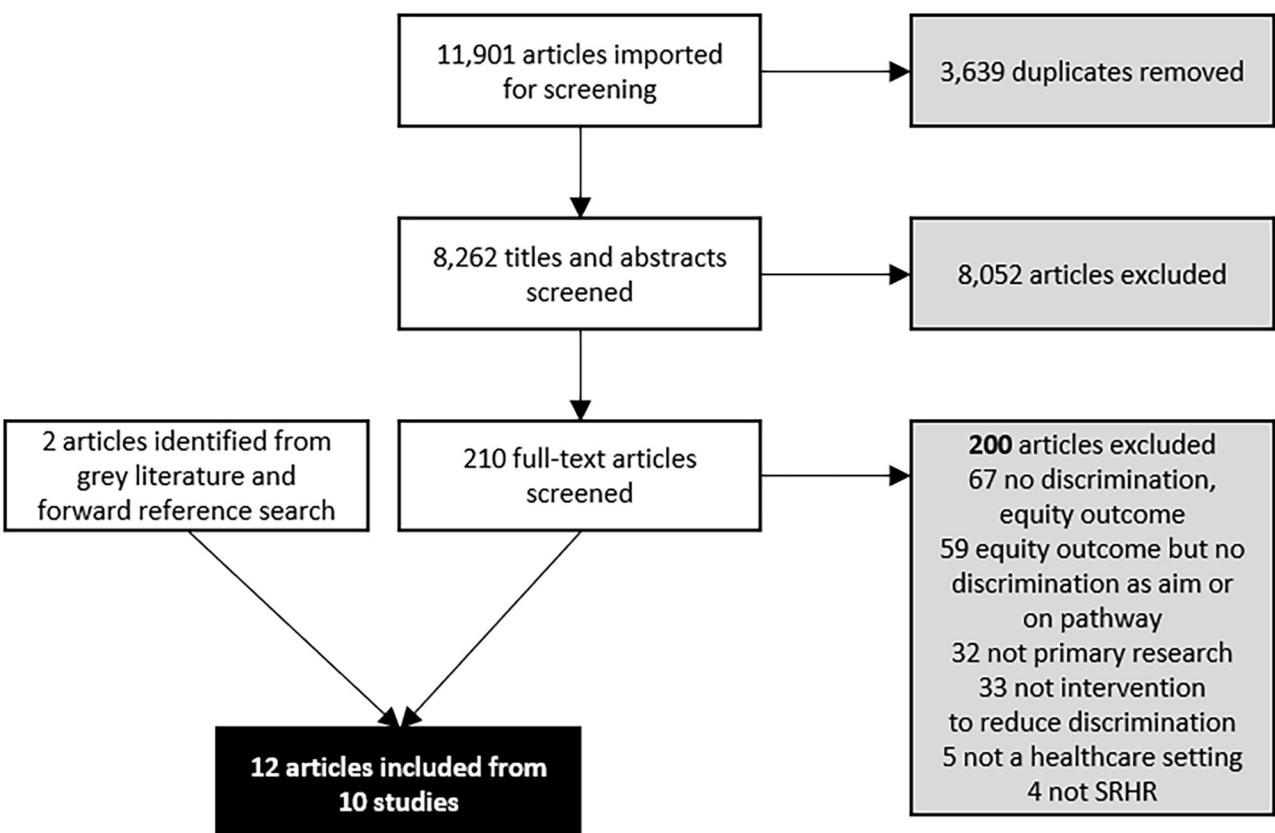

**Fig 2. PRISMA diagram.** This figure depicts the PRISMA flow diagram, detailing the database searches, number of abstracts and full texts screened, reasons for exclusion, and included studies and articles.

and HIV [60, 62], and one each on pregnant women with substance abuse [58], PMTCT [64], sexual health [59], sexual health, substance abuse, and STIs and blood-borne infections [61], and genitourinary medicine, family planning, and women's health [63]. Most articles (8/12) collected data from participants who were health and/or social workers [54–56, 58, 59, 61, 62, 65]; three articles included both health workers and healthcare users [60, 63, 64], and one article included only healthcare users (women who had abortions) [57]. Six of 12 included articles described only quantitative results [55, 58, 60, 61, 64, 65], three were qualitative evaluations of strategies [54, 56, 57], and three were mixed-methods evaluations of strategies [59, 62, 63]. Among the articles that included quantitative measures, one was a cluster randomized controlled trial [64], and the remaining articles used quasi-experimental designs, including six articles using a pre-test, post-test single group design [55, 59, 60, 62, 63, 65], and two articles used a post-test only design [58, 61]. We used the ROBINS-I tool to assess the risk of bias in the quantitative studies (S2 Table outlines detailed critical appraisal). Most (8/9) studies had an overall critical risk of bias, primarily due to bias from confounding (typically due to no control group or adjustment for potential confounders), selection of participants (typically due to self-selection into the intervention group), bias due to missing data (substantial loss to follow-up or blank surveys) and measurement of outcomes (typically because participants who knew they were in the intervention group reported the impact of the intervention on their knowledge or behavior). One study had a serious risk of bias due to deviations from intended interventions, missing data, and measurement of outcomes. We used an adaptation of the CASP

tool to assess methodological limitations in qualitative studies (S3 Table). These studies had minor to moderate concerns, due to limitations in data collection, reflexivity, and data analysis.

## Narrative synthesis

**Description of interventions.** Table 2 describes the characteristics of the included strategies, using an adaptation of Nyblade and colleagues approach for categorizing approaches to reduce stigma and discrimination [12]. Of the 10 strategies (Harris et al., 2011, Martin et al., 2014, and Debbink et al., 2016 evaluated the same strategy), most (7/10) provided information to healthcare providers about health conditions affecting the stigmatized population or discrimination experienced by this group [57–64]. Six out of 10 strategies aimed to build skills or improve the competence of providers working with a stigmatized group (as defined by the researchers) [58–64]. Four out of ten strategies were participatory and designed with or by the stigmatized group [55, 59, 61, 65]. Three strategies involved engagement or contact with the stigmatized group as part of the strategy [55, 59, 65]. Four strategies used empowerment approaches to help improve coping mechanisms to overcome stigma (such as empowering stigmatized groups–including abortion providers–with knowledge and skills) [55, 57, 64, 65], and three strategies included structural or policy reform (such as integrating health services or changing models of care) [59, 63, 64]. Half of the strategies had repeated components, such as multiple workshops (off-site or online, followed by on-site, or follow-up monitoring or technical assistance) [55, 59–61, 64], and the remaining were once-off workshops [57, 58, 62, 63, 65].

**Quantitative evaluation of impact.** Fig 3 describes the quantitative measurement approaches and outcome evaluation among the nine articles with quantitative outcomes. All articles measured health worker-reported outcomes [54, 55, 58–61, 63–65]. Three articles

| Author (year) | Providing information about health conditions or stigma? | Building skills or improve competence working with a stigmatized group? | Participatory: designed with or by stigmatized group? | Involve engagement or contact with stigmatized group? | Empowerment approach (improve coping mechanisms to overcome stigma? | Structural or policy reform? | Once-off or repeat intervention |
|---|---|---|---|---|---|---|---|
| Duby 2019 (61) | Yes | Yes | Unclear | No | No | No | Once-off |
| Geibel 2017 (59) | Yes | Yes | No | No | No | No | Repeat (2 workshops) |
| Jadwin-Cakmak 2020 (58) | Yes | Yes | Yes | Yes | No | Yes | Repeat (online, in-person and 3-month technical assistance) |
| Kinn 2003 (62) | No | No | No | No | No | Yes | Once-off (reorganization of services) |
| Littman 2009 (56) | Yes | No | Unclear (women shared abortion stories on DVD) | No | Yes | No | Once-off |
| Maclean 2018 (60) | Yes | Yes | Yes | No | No | No | Repeat (3 workshops) |
| Martin 2014, Harris 2011, Debbink 2016 (53-55) | No | No | Yes (with abortion providers, who were stigmatized) | Yes (with abortion providers, who were stigmatized) | Yes (with abortion providers, who were stigmatized) | No | Repeat (6 workshops) |
| Mosley 2020 (64) | No | No | Yes (with abortion providers, who were stigmatized) | Yes (with abortion providers, who were stigmatized) | Yes (with abortion providers, who were stigmatized) | No | Once-off (2-day retreat) |
| Phiri 2019 (63) | Yes | Yes | Unclear | No | Yes | Yes | Repeat (off-site & on-site training, ongoing monitoring) |
| Seybold 2014 (57) | Yes | Yes | No | No | No | No | Once-off |

**Fig 3. Characteristics of included interventions.** This figure presents the characteristics of included interventions, mapped to the six main strategies to reduce stigma identified by Nyblade and colleagues [45].

measured both health worker-reported and client-reported outcomes [60, 63, 64]. Four articles used validated measurement tools [55, 60, 64, 65]. One study also reported retention in care at 12 months for babies with mothers living with HIV [64].

Given the heterogeneity in outcome measurement (e.g. different tools, scales and measurement approaches) and evaluation (e.g. reporting percentages, means, odds ratios, and beta-coefficients), meta-analysis was not possible. Fig 3 narratively reports the impact of the interventions on the relevant outcomes of interest. In general, strategies that gave providers information about health conditions and/or stigma among the population group of interest reported increased self-reported awareness of the challenges faced by and decreased assumptions made about the population group. However, there were limitations in whether newfound knowledge was translated into changed practice or improved confidence in caring for population groups of interest, as these outcomes were typically not measured or reported by either providers or users. Of the three articles that evaluated user-experiences after the intervention [60, 63, 64], there were mixed results on the impact of the strategy on their experiences of care. In client exit interviews, Geibel and colleagues found that healthcare users were more likely to discuss being a member of a key population group and disclosing sexual activity to providers, and less likely to report the provider acted in a discriminatory way [60]. Phiri and colleagues found that women living with HIV were more likely to report social support and reduced perceived stigma from healthcare workers, despite no effect on retention of their babies in care at six or twelve months [64]. Kinn and colleagues found that service integration increased perceived confidentiality and happiness with overall care, but had no effect on the quality or availability of services [63]. Of note, the three articles that evaluated user experiences after the intervention collected data at two and six months after intervention [64], six months after the intervention [63], and six and twelve months after the intervention [60], which limits the evaluation of the sustainability of the intervention effects over time.

**Qualitative evaluation of impact.** Six articles contributed qualitative evidence (including qualitative evidence from three mixed-methods studies) related to stigma and discrimination in abortion care, sexual health, and service integration [54, 56, 57, 59, 62, 63]. One article with qualitative evidence following implementation was from the perspective of women who had had an abortion [57], four articles were from the perspective of health workers after implementation [54, 56, 59, 62], and one article was from the perspective of health workers before and after implementation [63].

Three articles described two strategies (introducing a culture of support for abortion patients [57] and the Providers Share Workshop [54, 56]) that aimed to address internalized stigma among women who had abortions and abortion care providers. Both strategies were viewed positively, and perceived by participants to be validating, supportive, and contributed to "feeling understood" and less alone [54, 56, 57]. While the strategies took different forms (reflexive workshops [54, 56], educational information in a film, and brochure [59]), participants reflected that the materials helped them to realize that they could openly discuss their feelings of stigmatization with others. Abortion providers felt that the workshops engendered an '*esprit de corps*' [collective spirit] that helped them to manage the consequences of feeling stigmatized through the safe exchange of ideas [54, 56].

Two articles described strategies that aimed to address perceived and enacted stigma among healthcare providers working with stigmatized groups (youth who identify as gender non-binary and/or do not identify as heterosexual) [59] and men who have sex with men, sex workers and people who use drugs [62]). Healthcare providers described that the interventions encouraged introspection and cultural humility to confront their own prejudices, and increased their empathy and compassion to the stigmatized groups, as well as improved their

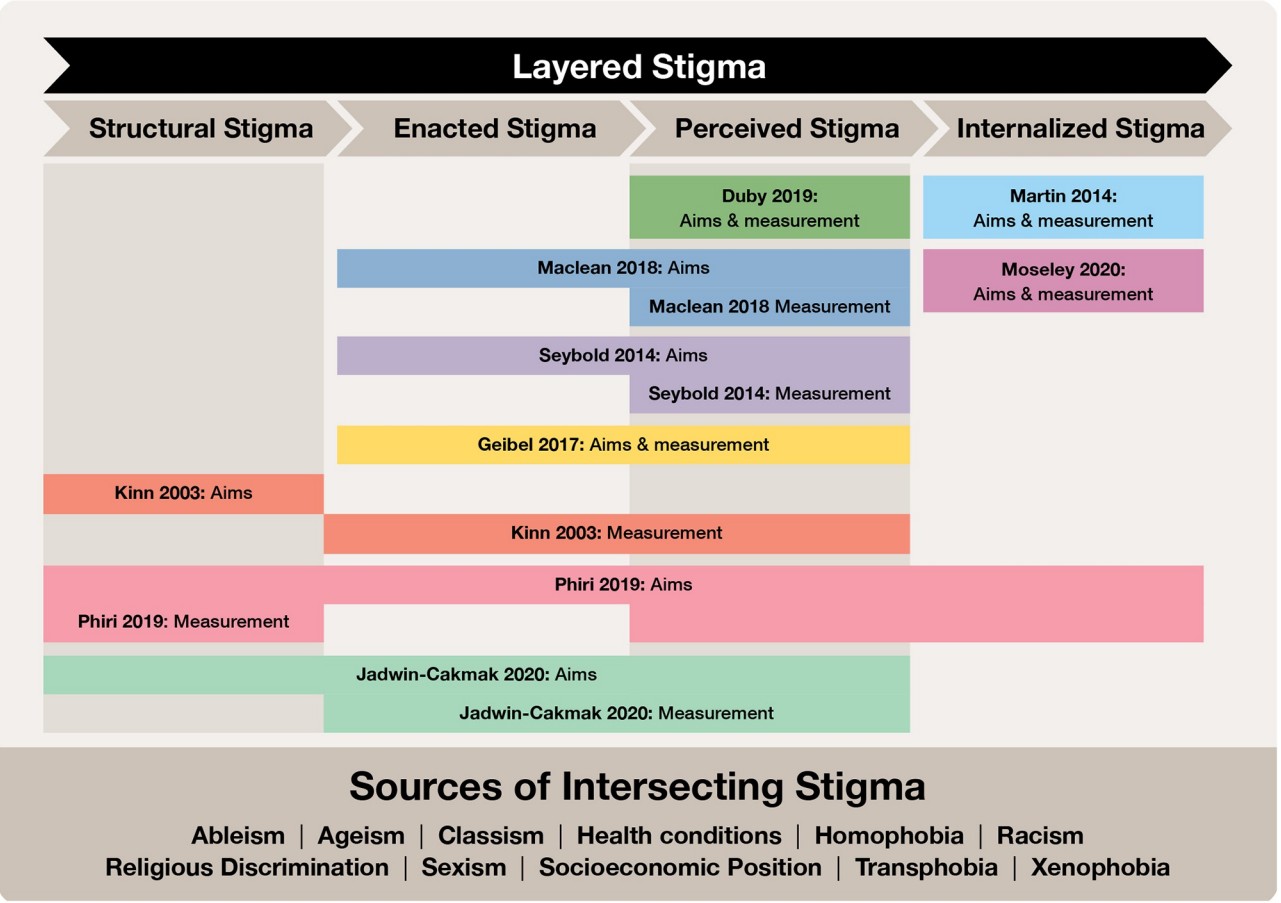

**Fig 4. Mapping included intervention aims and outcome measures using a multi-level stigma model.** This figure depicts a mapping of the nine studies with relevant quantitative outcomes on stigma or discrimination to the different levels at which stigma can exist and be confronted. First, we mapped the studies based on which level or levels of stigma (internalized, perceived, enacted, structural, layered) the strategies aimed to address and how they were measured (Tables 1 and 2). Where strategies aimed to address or measure across more than one level of stigma, we indicated as such. Then we visually mapped the strategy aims and measurement approaches across the different levels of stigma to explore where the aims and measurement approaches were similar and different.

understanding of stigma and marginalization [59, 62]. However, there was no discussion on whether their changed attitudes led to meaningful change in clinical encounters.

One study described the impact of a strategy to address structural stigma through the introduction of integrated services [63]. Healthcare providers felt that the integrated services reduced the stigma associated with attending sexual health services, and perceived the integration as beneficial to healthcare users [63].

**Mapping interventions and outcome measurement to the levels at which stigma exists and can be confronted.** Fig 4 maps the nine studies with relevant quantitative outcomes on stigma or discrimination to the different levels at which stigma can exist and be confronted. First, we mapped the studies based on which level or levels of stigma (internalized, perceived, enacted, structural, layered) the strategies aimed to address and how they were measured (Tables 1 and 2). Where strategies aimed to address or measure across more than one level of stigma, we indicated as such. Then we visually mapped the strategy aims and measurement approaches across the different levels of stigma to explore where the aims and measurement approaches were similar and different.

Four articles (4/9) aimed to intervene and measured within the same level of stigma [55, 60, 62, 65]: Martin et al. 2014 and Mosley et al. 2020 both aimed to address and measured internalized stigma among abortion providers; Duby et al. 2019 aimed to address and measure perceived stigma among health workers and service users (sexual health); and Geibel et al. 2017 aimed to address and measure perceived and enacted stigma among health workers and service users (youth SRHR).

Five articles (5/9) had differences in the levels of stigma that the strategy aimed to intervene on, and the measurement approaches used [58, 59, 61, 63, 64]. These articles all aimed to address higher levels of stigma (structural and/or enacted stigma), but measured more proximal levels of stigma. For instance, Kinn 2003 aimed to address structural stigma by reconfiguring health services and measured the impact of service reconfiguration on perceived and enacted stigma [63]. Similarly, Jadwin-Cakmak aimed to address structural, enacted, and perceived stigma in sexual healthcare for gender- and sexuality-diverse youth after training workshops, and measured the impact on enacted and perceived stigma [59].

## Discussion

We identified 12 articles from 10 studies with strategies to reduce stigma and discrimination in SRHR settings. Our review highlights several critical gaps and opportunities in the literature, which are discussed below, including: 1) focus on describing the problem of stigma and discrimination but not intervening to address it; 2) strategies that aim to improve equity, access, or quality of care for marginalized groups, but do not address stigma and discrimination, and 3) strategies that focus on healthcare users' and providers' internalized, perceived, or enacted stigma, but ignore the structural and societal conditions that perpetuate stigma and discrimination.

While there is a substantial body of literature describing and measuring various types of stigma and discrimination experienced by people seeking care for sexual and reproductive health [66–69], there is very limited interventional work to reduce stigma and discrimination. In our review, we were ultimately only able to identify 12 articles from 10 studies that met our criteria for intervening to address these factors for sexual and reproductive health care. Our findings highlight that despite strong descriptive evidence of stigma and discrimination in healthcare settings, we have only just begun to use that evidence to design and evaluate strategies to address stigma and discrimination (which is further supported by the years of publication of included studies 2003–2020). We hypothesize three potential reasons for this critical gap. First, stigma and discrimination may be perceived as particularly challenging to address as they touch on potentially deeply rooted biases, so people are hesitant to try. Second, it is possible that researchers and implementers do not have a good framework to think through what types of strategies are likely to have the most effect. Third, power dynamics in funding and health leadership may have historically limited research and funding to address the impact of stigma and discrimination in healthcare settings. People with lived and professional experiences of stigma and discrimination may often not be in the position to implement the change they envision. They are thus often compelled to design studies to educate those in power on the magnitude and effect of the problem. This is reflected in the preponderance of descriptive work, with limited interventional work. More research is urgently needed to develop and test different strategies to address the persistent problem of stigma and discrimination in healthcare settings.

Our findings are consistent with previous systematic reviews on strategies to address stigma and discrimination in maternal and child health, which found limited publications on the topic [5]. This may be because maternal and child health may comparatively not be considered

stigmatized health areas. In contrast, systematic reviews on strategies that focus on stigmatized health conditions tend to yield more publications; for example, Nyblade and colleagues' systematic review on stigma in health facilities identified forty-two studies focusing on strategies to reduce HIV, mental illness, and substance abuse stigma [45]. Most strategies focused on reducing HIV-related stigma in healthcare settings [12, 70]. However, given how much has been written on describing HIV-related stigma and discrimination, it is surprising that a recent systematic review of quantitative studies (with comparative designs) of strategies to reduce HIV-related stigma and discrimination in healthcare settings found only 14 eligible papers reporting on eight studies [70]. Nayar and colleagues note that even with the HIV literature, there are no published studies on the effect of stigma-reduction strategies for pregnant women living with HIV or the direct impact of HIV-related stigma on the uptake of PMTCT services [5]. Similarly, studies of strategies that target stigma and discrimination as a means to improve neonatal survival and health are nearly nonexistent, except in the PMTCT literature, where strategies have focused on eliminating barriers to care among pregnant women living with HIV [42]. The dearth of research exploring and evaluating the impacts of strategies to reduce stigma and discrimination in healthcare settings, using rigorous design and evaluation methods to measure impact, is a critical gap identified in our review.

Another critical gap we identified is how many strategies aim to improve the health and well-being of marginalized groups, but do not measure stigma and discrimination outcomes. For example, our initial screening yielded several publications based on the Janani Suraksha Yojana (JSY) scheme—a conditional cash transfer program in India that incentivizes women to give birth in a health facility as a way to reduce maternal and neonatal mortality [71, 72]. However, all of these studies were ultimately excluded because, although this program was intended to reduce inequities and improve access, none measured stigma or discrimination as outcomes. Other studies included interventions to improve access through voucher programs, or mobile or special clinics for marginalized groups [73–76], which likewise did not measure stigma and discrimination as outcomes. This is important to note because these interventions, although well intended, could further lead to stigmatization of marginalized groups if not designed intentionally to reduce stigma and discrimination.

We also identified a lack of a comprehensive examination of specific types of stigma and multiple stigmas experienced. Most studies included in our review focused on internalized, perceived, or enacted stigma. While three studies attempted to address structural stigma [59, 63, 64], including through the integration of health services or changing models of care, only one study actually measured structural stigma [64]. This highlights the challenges in the measurement of different levels of stigma, particularly structural and layered stigma, which operates at higher and across multiple levels. While it is critical to identify specific strategies to address individual levels of stigma, a layered approach is needed to truly address health equity and to recognize the intersectional oppressions at play across multiple marginalized and stigmatized groups. Moreover, focusing on lower-level stigmas, such as internalized stigma or perceived stigma, places responsibility on individuals (healthcare users and providers) to address issues of stigma and discrimination, and overlooks systemic and organizational drivers. Indeed, most strategies focused on educating providers while others involved healthcare users themselves in the development of strategies. While centering user experiences is important to develop culturally-appropriate strategies, health system management and policymakers also have an important role to play in addressing underlying societal power inequities [77]. Most studies focused on stigmas related to health conditions—such as abortion stigma or SRH stigma—failing to highlight the underlying social conditions that drive stigma and discrimination, such as classism, racism, ableism, and xenophobia. The patient-provider relationship, as well as the experiences of healthcare users are, however, often a reflection of deeper dynamics

of power inequities in societies in which they are embedded [77]. This includes the subordination of certain racial groups, women's subordinate position to men, and the marginalization of specific populations such as LGBTQ+, those with disabilities, and so on. A greater emphasis on how to address stigma and discrimination that stems from inequities in health systems, social and economic policies, community social norms, and power differentials at the societal level is critical in addressing stigma and discrimination in the healthcare setting.

## Strengths and limitations

Our review has several limitations. We excluded strategies that aimed to increase access to health services (such as policy-level abortion reform, and increasing access to facility-based birth), unless stigma or discrimination was an outcome measure. While this may have excluded some structural interventions that could potentially address discrimination related to societal conditions that constrain opportunity or well-being, we note that increasing access alone does not necessarily equate to a reduction in stigma or discrimination. We also excluded strategies that did not aim to reduce stigma or discrimination, but may have unintentionally reduced stigma or discrimination, for example, strategies to improve person-centered care. Given the complexity of addressing stigma, we believe that effective strategies should explicitly aim to reduce stigma and discrimination and measure the impact of the strategy on stigma and discrimination-related outcomes. Our review also has several notable strengths. Finally, we explored how we could assess the differences in intervention effects based on the "doses" of the intervention (once-off, or repeat). However, we found that due to the heterogeneity in outcomes reported, and multiple (and often many) outcomes reported in the quantitative studies, we are unable to conclude anything meaningful about the impact of a once-off versus repeat training on outcomes of interest. We hypothesize that given the complexity of addressing stigma and discrimination, a once-off training would be less likely to evoke change (especially sustainable change), and that measurement of this in the included studies is limited by evaluation of intervention impact typically occurring close to the time of the intervention. This may be particularly true for interventions targeting areas other than increasing provider awareness, which we hypothesize would be likely to need repeat and sustained engagement with stakeholders to influence systems change.

We used a systematic review approach, which increases the credibility, reliability, and transparency of the findings. Our broad inclusion criteria for types of strategies and types of studies (quantitative, qualitative, and mixed-methods) also ensured we captured the range of studies relevant to the topic. Our international team of researchers has broad and complementary experience across public health, clinical practice, and social and reproductive justice that enriched the quality and scope of the review.

## Implications for policy and practice

The findings from this review have important implications for policy and practice. Individuals who experience stigma and discrimination when accessing sexual and reproductive health care are at risk of harm and poor health outcomes. But beyond that, it is a violation of their human rights. Thus, strategies are urgently needed to eliminate stigma and discrimination in SRHR settings. In designing, implementing, monitoring, and evaluating interventions to reduce stigma and discrimination, it is essential that the views of the group or community experiencing stigma and discrimination are incorporated. Ideally, such interventions should be led or co-led by representatives from the particular group or community [78], to ensure that their needs and experiences are addressed. We note that while the inspiration for this review stemmed from our work on mistreatment during childbirth and respectful care, we did not

identify any strategies to promote respectful care or reduce mistreatment during childbirth that explicitly included strategies to eliminate stigma and discrimination. Moving forward, we encourage researchers, programmers, and policy-makers to ensure that this critical component of respectful care is not neglected.

Health care providers need to be supported to undertake appropriate education and training to understand and overcome discriminatory care practices. However, training health care providers in and of itself is not enough to drive change [8]. Health services need to enact structural and policy reform to create safe, inclusive, and respectful environments for people both providing and accessing SRH services. In some instances, stigma and discrimination can affect both the healthcare users and providers of sexual and reproductive health care, as can occur with abortion care [79]. As such, it is likely that effective interventions will be complex, operating across multiple levels to maximize the likelihood for change.

Interventions that address stigma and discrimination across all layers, especially at the enacted and structural levels are lacking. It is at this societal and political level where enabling legal and policy contexts and social norms can strongly influence the care provided and received by people accessing SRH services. Efforts to reduce legal restrictions and drive broader social and cultural change can have an impact at an individual, community, and institution level. Activists can also play a major role in not only encouraging community awareness, but also policy change. These broader societal, cultural, and legal shifts can have a significant influence on health care providers, influencing decision making, referral processes, and care provided.

## Conclusions

While important foundational work has been done to describe and define stigma and discrimination in sexual and reproductive healthcare settings, more work is urgently needed to intervene to eliminate stigma and discrimination in these settings, to achieve respectful, person-centered, and equitable care for all. Moreover, healthcare and policy interventions that aim to improve equity should consider measuring stigma and discrimination-related outcomes, as equity cannot be achieved when stigma and discrimination persist. While some work has been done to address perceived and enacted stigma, more work is needed to address structural and layered stigma to challenge and dismantle the societal conditions, sociocultural norms, and institutional policies that influence the opportunities and well-being of stigmatized groups. Provision of sexual and reproductive healthcare free of stigma and discrimination is a basic and essential tenet of every health system. The Sustainable Development Goals related to sexual and reproductive health targets will therefore not be achieved until health systems ensure people have access to sexual and reproductive healthcare free of stigma and discrimination.

## Supporting information

**S1 Table. PRISMA checklist.**
(PDF)

**S2 Table. Risk of bias assessments for quantitative studies (ROBINS-I).**
(PDF)

**S3 Table. Critical appraisal of qualitative studies (CASP).**
(PDF)

**S1 Text. Review protocol.**
(PDF)

**S2 Text. Search strategies: Medline, CINAHL, Global Health.**
(PDF)

## Acknowledgments

Thank you to Rio With all for their support in designing the multi-level stigma models in Figs 1 and 3, and to Patrick Condon for his assistance with the search update.

We appreciate the valuable contributions from the WHO working group on interventions to reduce mistreatment of women during childbirth: Gita Sen, Aditi Iyer, Bhavya Reddy, Sophia Thomas, Ravi Sadhu, Klaartje Olde Loohuis, Joyce Browne, Winter Bruner, Annemoon Jonker, Emmanuela Salia, Hannah Brown Amoakoh, Mary Amoakoh-Coleman, Bregje de Kok, Sasha Kruger, Rick Grobbee, Özge Tunçalp, Hedieh Mehrtash, Anayda Portela, Emmanuel Srofeneyoh, Kwame Adu Bonsaffoh, Soo Downe, Rebecca Nowland, Alan Farrier, Cath Harris, Lisa Tanyaradzwa Gondo, Kenny Finlayson, Gill Thomson, Carol Kingdon, Andy Clegg, Lynn Freedman, Marta Schaaf, and Maayan Jaffe.

## Author Contributions

**Conceptualization:** Özge Tunçalp, Patience A. Afulani.

**Data curation:** Meghan A. Bohren, Jim Berryman.

**Formal analysis:** Meghan A. Bohren, Martha Vazquez Corona, Osamuedeme J. Odiase, Alyce N. Wilson, May Sudhinaraset, Nadia Diamond-Smith, Patience A. Afulani.

**Funding acquisition:** Meghan A. Bohren, Özge Tunçalp, Patience A. Afulani.

**Investigation:** Meghan A. Bohren, Martha Vazquez Corona, Osamuedeme J. Odiase, Alyce N. Wilson, May Sudhinaraset, Nadia Diamond-Smith, Patience A. Afulani.

**Methodology:** Meghan A. Bohren, Martha Vazquez Corona, Osamuedeme J. Odiase, Alyce N. Wilson, May Sudhinaraset, Nadia Diamond-Smith, Patience A. Afulani.

**Project administration:** Meghan A. Bohren.

**Visualization:** Meghan A. Bohren.

**Writing – original draft:** Meghan A. Bohren, Patience A. Afulani.

**Writing – review & editing:** Meghan A. Bohren, Martha Vazquez Corona, Osamuedeme J. Odiase, Alyce N. Wilson, May Sudhinaraset, Nadia Diamond-Smith, Jim Berryman, Özge Tunçalp, Patience A. Afulani.

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
