## [Decision Letter · Decision Letter 0]

10 Mar 2022

PGPH-D-21-00384

Strategies to reduce stigma and discrimination in sexual and reproductive healthcare settings: a mixed-methods systematic review

Dear Dr. Bohren,

Thank you for submitting your manuscript to PLOS Global Public Health. After careful consideration, we feel that it has merit but does not fully meet PLOS Global Public Health’s publication criteria as it currently stands. Therefore, we invite you to submit a revised version of the manuscript that addresses the points raised during the review process. I apologize for the lengthy review process, but hope that the authors find the feedback useful.

As you will see from the reviewers' comments, this manuscript has potential for impact and important contributions to the SRHR literature.Please address the reviewers' minor comments on wording and clarifications sought, which will strengthen the article's utility to the literature. I agree with the reviewers' comments and do not have additional input to add.

We look forward to receiving your revised manuscript.

Kind regards,

Marie A. Brault, PhD

Academic Editor

Journal Requirements:

1. Please update your Competing Interests statement. If you have no competing interests to declare, please state: “The authors have declared that no competing interests exist.”

2. Please provide separate figure files in .tif or .eps format only and ensure that all files are under our size limit of 20MB.

Reviewers' comments:

Reviewer's Responses to Questions

**Comments to the Author**

1. Does this manuscript meet PLOS Global Public Health’s publication criteria? Is the manuscript technically sound, and do the data support the conclusions? The manuscript must describe methodologically and ethically rigorous research with conclusions that are appropriately drawn based on the data presented.

Reviewer #1: Yes

Reviewer #2: Yes

2. Has the statistical analysis been performed appropriately and rigorously?

Reviewer #1: N/A

Reviewer #2: N/A

3. Have the authors made all data underlying the findings in their manuscript fully available (please refer to the Data Availability Statement at the start of the manuscript PDF file)?

Reviewer #1: Yes

Reviewer #2: Yes

4. Is the manuscript presented in an intelligible fashion and written in standard English?

Reviewer #1: Yes

Reviewer #2: Yes

5. Review Comments to the Author

Reviewer #1: General comment: Thank you for the opportunity to read and review this paper. The authors are to be commended for this systematic review of interventions designed to reduce stigma and discrimination in sexual and reproductive healthcare settings. This paper is of international public health relevance, all people have the right to access SRH services that are free from stigma and discrimination which have significant negative impacts on the SRH and wellbeing of individuals in all parts of the world.

The authors set out to review the evidence for strategies and interventions that aim to reduce stigma and or discrimination in sexual and reproductive health care settings, taking into account the multiple levels on which stigma operates. They used a narrative synthesis approach which is appropriate for the topic area with both quantitative and qualitative studies included in the review. No meta-analyses or other statistical analyses were undertaken due to the heterogeneity of outcome measurement in studies reviewed. Having identified 7000+ papers, only 12 met inclusion criteria (9 included quantitative aspects and 3 qualitative) - which is a clear indicator that more work is urgently needed in this area.

The authors have clearly summarized the findings of included studies, and also highlight the gaps and opportunities for future research. They have suggested the ways in which future researchers could consider different types of stigma and highlight the importance of involving the communities of interest in the design of such work. Another important message is that designing interventions to improve outcomes for marginalized groups without specific consideration of stigma and discrimination or measuring these, is unlikely to succeed.

INTRODUCTION

Comment: This provides an informative and well-written description of the relevant terms, impacts and understandings of stigma and discrimination as they relate to SRH services and outcomes. Referencing is thorough and current.

METHODS

Comment: The methods are well described and in enough detail for others to follow or replicate. Information recommended for inclusion in the PRISMA checklist has been included in sufficient detail. The risk of bias has been acknowledged and discussed in the paper and further details included in S4 appendix. The study protocol has been provided and appears to align with the work undertaken.

Minor queries:

1. P8, lines 188-190. ‘We excluded STI and HIV counseling, testing, and treatment services if conducted outside the context of contraceptive or maternal health services, as these topics have been well documented in the literature (15, 27, 28). Can the authors please clarify this sentence reads as intended – many STI and HIV testing and treatment services would be separate to contraceptive or maternal health services and I’m not sure that we already know how to address/reduce stigma and discrimination for these aspects of SRH. Refs 27 and 28 are about HIV-related stigma, whereas Nyblade et al are looking at stigma across a wide range of health conditions including HIV but not SRH specifically.

2. It isn’t immediately clear why S5 appendix includes only two of the qualitative studies – does the table just include studies where concerns about biases arose (whereas the other studies were assessed and deemed free from biases?) Please clarify this for readers.

RESULTS

Comment: Very good use of tables and figures to summarize and collate the review findings, all aspects in the PRISMA checklist relating to results are covered. Figure 1 is excellent and Figure 4 is a really effective way to show what type(s) of stigma the different studies attempt to address.

Minor suggestions:

3. Tables 1 and 2 – It would be helpful to have the reference numbers included alongside author names to make it easier for readers to match up the written summary of studies (on p11, and p 16) with the details in the tables

4. Figure 3 – Please add in reference numbers alongside Author/year.

DISCUSSION

Key findings are well summarized and implications for future research discussed. Many studies have described stigma and discrimination related to SRH, but few studies have attempted to address this challenging topic. This paper will help other researchers to conceptualize the way(s) in which they might go about setting up an intervention to tackle one or more levels of SRH-related stigma and discrimination.

Reviewer #2: Thank you for the opportunity to review this manuscript which reports on a mixed-methods systematic review of strategies to reduce stigma and discrimination in healthcare settings offering sexual and reproductive care.

My comments include suggestions for some textual modifications in the manuscript and methodological and analytical queries regarding the data and findings presented.

Abstract

Reads very well.

Introduction

Page 3: Lines 8-9

Suggest re-phrasing as “The 2030 Sustainable Development Agenda highlights two

targets specific to achieving universal access to sexual and reproductive health, health-care services and reproductive rights”. As presently phrased, it might be understood that the SDG targets also encompasses “sexual rights”, which isn’t the case.

The introduction and brief discussion of “Reproductive Justice” as a discourse that identifies the intersecting structural factors contributing to inequities in SRHR outcomes can be better contextualised for the reader and its relevance to the topic of this manuscript more clearly spelt out. For example, the international community is likely very familiar with the discourse on human rights, hence providing some background on why or how an RJ approach extends or builds upon or addresses the limitations of a human rights framework to achieving SRHR would be beneficial information to include. I do think the discussion on RJ would be better placed once an overview of the various types of stigma are presented (Figure 1). RJ can be a useful lens through which to understand how structural stigma can emerge through the intersecting influences of law and social policy that have contributed to longstanding structural oppression and injustice. Doing so, can guide the reader to look beyond the inter-personal aspects of stigma and implicating the institutions and systems that perpetuate it at a structural level.

Stigma and discrimination in healthcare settings

Page 4: Lines 51-52

The phrase “socially undesirable” in the explanation for Enacted stigma is confusing. Instead suggest re-phrasing as “Enacted stigma refers to the manifestations of unfair treatment arising from adverse social judgment”

Figure 1

Page 34

This is a useful visual to depict the various types of stigma and how they manifest in individuals, the health system and society. A few related queries and suggestions:

Within “Perceived Stigma”, it is unclear to me how the following three components are conceptually distinct to necessitate separate labels:

1. “Healthcare provider perspectives of individual or group attributes (e.g., gender identity, homophobia, racism)”

2. “Healthcare provider perspectives and internalized belief systems of individuals or groups”

3. “Internalised belief systems”

Instead, I would suggest combining the above into the following: “Healthcare provider perspectives and belief systems regarding individual or group attributes “(e.g., related to gender identity, sexual orientation, racial group etc.,).

I would also recommend having “Internalised belief systems” as a stand-alone component of “Internalised Stigma”, as you indicate in the text that this form of stigma is applied by the individual onto themselves, which is likely related to their own belief systems, shaped by societal norms and expectation.

“Labelling, stereotyping, othering” which are the processes by which stigma is manifested seems to fit better in “Enacted Stigma” than “Perceived Stigma”

Within “Enacted Stigma” rather than having “poor person-centered care” as a component with examples, I would include the examples themselves as separate components within this domain as they are sufficiently descriptive to indicate that these actions constitute poor person-centered care. If the authors decide to keep “poor person-centered care” I suggest at least having “Denial of care” as a separate component since this is conceptually different from the provision of sub-standard care, as the latter assumes that some type of care is provided.

In Structural Stigma, “Restrictive access to health services (e.g., abortion, contraception, fertility)” is repeated twice.

Strategies to reduce stigma and discrimination in healthcare settings

Line 127: Add colon after follows:

Types of studies

Page 7, lines 158-161.

In describing their operational definition of “strategy”, the authors combine aspects of the study design (e.g., mystery client methodology, non-randomised designs) with the strategies themselves (e.g., workshops, training, simulation etc). This is confusing because I view the two components as distinct. The earlier sentence in this paragraph (156-158) explains the type of study designs that were considered eligible for inclusion. In the operational definition of strategy, I would recommend that the authors focus on what type of strategies were considered eligible, i.e., policy reform, healthcare provider training, community-based approaches targeting healthcare users etc.

Topics of interest, types of participants, and settings

Page 8, Lines 178-187.

I commend the authors for using the Guttmacher-Lancet Commission’s definition of SRHR. Included in the Commission’s definition of the package of SRHR services but missing from the manuscript are (a) prevention, identification, and counselling for gender-based violence (outside of maternal health programs), (b) comprehensive sexuality education, and (c) treatment of infertility. Can the authors comment on why this is the case, and include this in the text?

Data management, analysis and synthesis

Lines 241-245

In describing the approach to narrative synthesis, the authors discuss evaluating how “critical design factors” and “relationships between study designs” influenced the outcomes of interest. Can the authors clarify here and, in the text, whether this refers to the design of the study itself (e.g., quasi-experimental designs, RCTs, qualitative study etc.,) or the design of the intervention or both?

Description of interventions

Page 16, lines 28-30

The authors note that one-half of the interventions included repeated components whereas the remaining were once-off sessions; were there any noticeable differences in the impact based on differential “doses” of the intervention? The authors may want to comment on any differences or lack thereof in the text.

Quantitative evaluation of impact

Page 19, lines 50-51

The authors mention that there are “limitations” in whether information-based strategies for stigma reduction translated into changed practice or increased confidence in care. Could these limitations be described in brief?

I commend the authors in conducting a risk of bias assessment for the quantitative studies but would also like the authors to comment in this section whether any aspects of the “serious” and “critical” assessments of bias for all included studies are important to note in the context of how “valid” the reported results are likely to be, especially for those studies where an “effect” of the intervention is reported.

Discussion

Thorough and well-written.

6. PLOS authors have the option to publish the peer review history of their article (what does this mean?). If published, this will include your full peer review and any attached files.

**Do you want your identity to be public for this peer review?** For information about this choice, including consent withdrawal, please see our Privacy Policy.

Reviewer #1: No

Reviewer #2: No

---

## [Editor Report · Decision Letter 1]

13 May 2022

Strategies to reduce stigma and discrimination in sexual and reproductive healthcare settings: a mixed-methods systematic review

PGPH-D-21-00384R1

Dear Dr. Bohren,

We are pleased to inform you that your manuscript 'Strategies to reduce stigma and discrimination in sexual and reproductive healthcare settings: a mixed-methods systematic review' has been provisionally accepted for publication in PLOS Global Public Health.

Best regards,

Marie A. Brault

Academic Editor

I appreciate the thorough responses to the reviewers. All comments have been addressed.